

# State monitoring and fault prediction of centrifugal compressors based on long short–term memory and principal component analysis (LSTM-PCA)

Yuan Wang and Shaolin Hu

Automation School, Guangdong University of Petrochemical Technology, Maoming, Guangdong, China

## ABSTRACT

Centrifugal compressors are widely used in the petroleum and natural gas industry for gas compression, reinjection, and transportation. Early fault identification and fault evolution prediction for centrifugal compressors can improve equipment safety and reduce maintenance and operating costs. This article proposes a dynamic process monitoring method for centrifugal compressors based on long short-term memory (LSTM) and principal component analysis (PCA). This method constructs a sliding window for monitoring at each sampling point, which contains 100 data from the past and current time points, and uses LSTM to predict 30 future data points. At the same time, this method is also combined with the PCA threshold process monitoring method to construct a new LSTM-PCA monitoring algorithm. And the method was validated using centrifugal compressor process data. The results show that this method can effectively detect process anomalies, The improvements significantly reduced the false positive rate of detected anomalies, and can make multi-step advance predictions of system behavior after faults occur.

## INTRODUCTION

As an indispensable core equipment in the petrochemical industry, the centrifugal compressor's ability to operate stably at rated power is directly related to the quality of the product, the economic benefits of the company, and the safety of personnel. In actual production activities, it is difficult to avoid system failures due to internal factors (parts damage) or environmental factors (such as ventilation, sealing). The compressor system consists of a lubricating oil system, cooling system, power system, etc. Once these complex systems, which are composed of many units with different relationships, malfunction, it will usually affect the normal operation of the system and even cause immeasurable losses. Although the equipment can maintain short-term operation in a critical safety state, the potential risk of mechanical failure is still huge. Once a failure occurs, it may deteriorate rapidly, causing the system to fail to be repaired in time and causing serious problems. In addition, traditional monitoring systems are susceptible to

Corresponding author
Shaolin Hu, hfkth@126.com

data outliers, resulting in frequent false alarms, further increasing the complexity and cost of operation and maintenance. Therefore, how to accurately and timely monitor the status of centrifugal compressors has become a key issue to ensure the smooth progress of the production process and improve the level of safe production.

In recent years, with the rapid development of artificial intelligence and deep learning technology, research on compressor fault diagnosis has made significant progress, providing new solutions for equipment monitoring under complex working conditions. For example, multi-segmented attention-based long short-term memory networks (MA-LSTMs) are used for reciprocating compressor operation diagnostics on offshore oil platforms (*Tian & Li, 2022*). A composite model combining a deep convolutional autoencoder and a balanced sparse sampling informer (MCA-BS Informer) is used to solve the problem of long-term accurate health monitoring of compressors (*Tian, Ju & Feng, 2023*). An optimized probabilistic signal reconstruction method is proposed to address these challenges in fault prediction of rotating electrical machines using multivariable vibration signals (*Jiang et al., 2022*). Meanwhile, hybrid multimodal machine learning strategies are used to develop a data-driven system health monitoring framework (*Shen & Khorasani, 2020*). Artificial neural network (ANN) technology has successfully identified degradation of gas turbine engines due to erosion and fouling (*Giorgi, Ficarella & Carlo, 2019*). Self-powered fault diagnosis systems based on vibration energy harvesting have also achieved remarkable results (*Sato et al., 2022*). Classical dimensionality reduction methods, such as principal component analysis (PCA) (*Cheng, Xianwen & Yuan, 2020*), Independent Component Analysis (ICA) (*Yous & Hung, 2020*), and partial least squares analysis (PLS) (*Aminu & Ahmad, 2023*), have been widely used in compressor monitoring. Application and classification of PCA for troubleshooting the installation of MABs in petrochemical plant process facilities. Analyze root cause of failures using degree of variation and average variable threshold limits (*Shahid et al., 2024*). Combining multiscale principal component analysis with signature-based directed graph methods for process monitoring and fault diagnosis (*Ali et al., 2022*). Threshold recurrence graph-based and texture analysis applied to diagnose stiction in process control loops (*Kok et al., 2022*). A multi kernel support vector machine (MK-SVM) algorithm has been proposed to diagnose simultaneous faults in distillation columns (*Taqvi et al., 2022*). Combine multivariate exponentially weighted moving average (MEWMA) monitoring schemes with PCA modeling to improve anomaly detection performance (*Harrou et al., 2016*). Single-class support vector machine (KPCA-OCSVM) based on kernel principal component analysis and various kernels to learn an anomaly-free training set, then classify the test set (*Cheng et al., 2019*), among others. In addition, techniques based on big data machine learning, such as k-means clustering analysis based on big data machine learning of fault cases, study fault identification of rotating machinery without the support of external experts (*Wang et al., 2020*); The application of modular ensemble deep neural network (CC-MIDNN) for predicting flight data based on cluster clustering has also achieved good results (*Deng, Li & Zhao, 2024*). However, existing methods still have shortcomings in processing multi-dimensional feature data, dealing with noise interference, and reducing false alarm rates, and cannot fully meet the needs in actual

production. Therefore, further research and improvement of compressor fault detection technology have important academic value and practical application significance.

In order to solve the above problems, this paper proposes a monitoring method combining long short-term memory network (LSTM) and PCA. This method can not only capture the complex correlation of different characteristic data of the centrifugal compressor during operation, but also effectively deal with the interference of noise and abnormal values on the monitoring system, improving the accuracy of fault warning and the reliability of monitoring.

Specifically, the method in this paper comprehensively considers the past and future state change trends at each sampling point, making it possible to review the previous operating state and predict possible future changes. The time series correlation of the data is captured through the LSTM network, and the PCA dimensionality reduction is used to reduce the feature redundancy, which effectively improves the computing efficiency and monitoring ability of the model. This innovative method can significantly reduce the false alarm rate, especially under complex working conditions, identify potential failure risks more accurately and ensure the stable operation of the compressor system.

Through the application of this method, this article not only solves the unavoidable false alarm problem of traditional monitoring systems in critical safety states, but also provides enterprises with a more reliable equipment status monitoring solution, further ensuring the safe production and safety of the petrochemical industry.

In summary, the main contributions of this study are as follows:

(1) Based on the sample data obtained through sliding window processing, multi-step forecasting of data with multiple features was implemented.

(2) At each monitoring point, we have constructed a monitoring indicator that integrates both forecasted and historical data, and introduced a monitoring method based on LSTM-PCA. This method not only displays the past trends at each sampling point but also predicts future trends, thereby significantly enhancing the information content at each monitoring point.

(3) The method was analyzed and validated using process data from centrifugal compressors. The results demonstrate that this method can significantly improve issues related to false detections and critical safety insecurities.

## LSTM-PCA ALGORITHM DESIGN AND MODELING

In this paper, a fault-tolerant filtering algorithm (*Shaolin, Na & Wenming, 2016*) is used to filter and denoise the monitoring indicators. At the same time, in view of the large amount of data, the number of variables, and the strong coupling between the variables, principal component analysis is used to reduce the dimensionality of the data to shorten the training or optimization time of the model while ensuring the accuracy. Considering that monitoring data may be near critical states or contain anomalies, this study has designed a monitoring indicator for each monitoring point, which combines forecasted and historical data. By employing an improved LSTM-PCA algorithm, we are able to achieve real-time monitoring and fault prediction for compressors, thereby enhancing the accuracy and

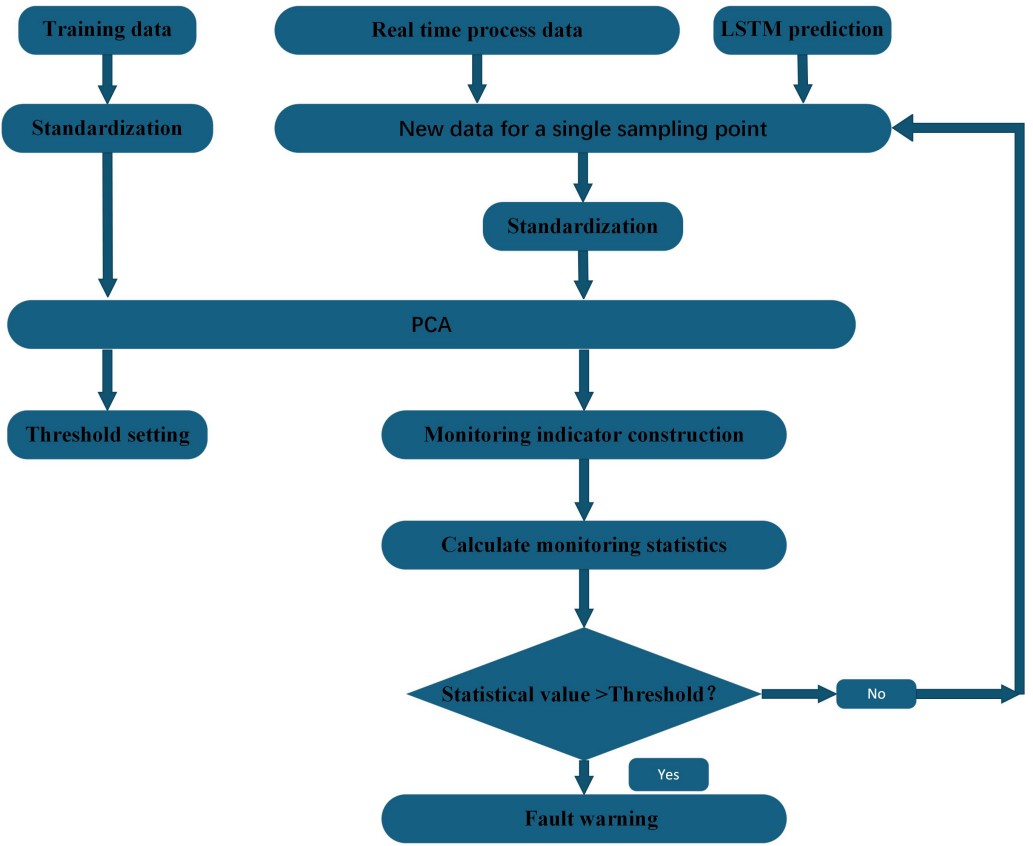

**Figure 1   Compressor condition monitoring and fault prediction structure diagram.**

reliability of fault detection. Figure 1 shows the condition monitoring and fault prediction structure of LSTM-PCA-based compressor proposed in this paper.

## Construction of monitoring indicators and threshold setting

In this paper, the training sample is 1,500 samples, and the monitoring sample window is a large sample of 131 data, and the two samples are independent of each other and conform to the normal distribution. We use a PCA-based process monitoring method to make judgments using the Hotelling-statistic and the square prediction error (SPE) statistic (*Bakdi & Kouadri, 2017*). The process is as follows:

Build a sampling matrix $Y_t$ with future and past data at each sampling point.

$Y_t$ by sliding window for 100 past and current moment data fragments $\{x(t_i)...x(t_{i-100})\}$ and 30 predicted snippets of future data $\{y(t_i)...y(t_{i+30})\}$ Compose.

$$Y_t = \begin{pmatrix} x_1(t_{i-100}) & x_2(t_{i-100}) & \cdots & x_{12}(t_{i-100}) \\ \vdots & \vdots & \ddots & \vdots \\ x_1(t_{i-1}) & x_2(t_{i-1}) & \cdots & x_{12}(t_{i-1}) \\ x_1(t_i) & x_2(t_i) & \cdots & x_{12}(t_i) \\ y_1(t_{i+1}) & y_2(t_{i+1}) & \cdots & y_{12}(t_{i+1}) \\ \vdots & \vdots & \ddots & \vdots \\ y_1(t_{i+30}) & y_2(t_{i+30}) & \cdots & y_{12}(t_{i+30}) \end{pmatrix}$$

where $x(t_i)...x(t_{i-100})$ are the 100 historical data before the current moment and $y(t_{i+1})...y(t_{i+30})$ are the 30 future data predicted at the current moment.

Then, the data is filtered by fault-tolerant filtering (*Shaolin, Na & Wenming, 2016*) to obtain the matrix $Y_t'$. Secondly, the data is normalized. The mean variance uses the mean variance of the training data, and the data in each column of the matrix $Y_t'$ is normalized to data with a mean value of 0 and a root mean square of 1, and the matrix $\hat{Y}_t$ is obtained.

Firstly, according to the offline data, the normal data $H$ is selected, and the $H$ is $\{H(t_1)...H(t_{1500})\}$, and the dimension is the same as that of the monitoring matrix. Then, the matrix is filtered for fault tolerance (*Shaolin, Na & Wenming, 2016*), and normalized to obtain the matrix $\tilde{H}$ The matrix $\hat{X}$ is obtained by normalizing the data in each column of the matrix $\tilde{H}$ to data with a mean value of 0 and a root mean square of 1.

Calculate the covariance matrices of the filtered and normalized training data and monitoring data respectively:

$$R_{ta} = \frac{1}{n-1}\hat{X}^T\hat{X} \tag{1}$$

$$R_{ts} = \frac{1}{h-1}\hat{Y}_t^T\hat{Y}_t \tag{2}$$

where n is the number of training samples, h is the number of monitoring samples, and the resulting covariance matrix is a feature 12*12-dimensional matrix.

The eigenvalues were sorted from large to small, and the top k features with a sum of more than 85% of the eigenvalues were selected for PCA dimensionality reduction, which was the eigenvalue.

$$\frac{\sum_{i=1}^{k}\lambda_i}{\sum_{i=1}^{m}\lambda_i} \geq 0.85. \tag{3}$$

Let the first k eigenvalues from large to small form a diagonal matrix, and k corresponding eigenvectors form a matrix. Namely:

The diagonal matrix and the corresponding eigenvectors of the training set form a matrix:

$$S_{k_1*k_1} = diag\left(\lambda_1, \lambda_2, \ldots, \lambda_{k_1}\right)$$

$$P_{m*k_1} = [p_1, p_2, \ldots, p_{k_1}]. \tag{4}$$

The diagonal matrix of the monitoring set and the corresponding eigenvectors form the matrix:

$$\hat{S}_{k_2*k_2} = diag\left(\hat{\lambda}_1, \hat{\lambda}_2, \ldots, \hat{\lambda}_{k_2}\right)$$
$$\hat{P}_{m*k_2} = [\hat{p}_1, \hat{p}_2, \ldots, \hat{p}_{k_2}]. \tag{5}$$

For $\hat{X}, \hat{Y}_t$, the number of samples is still n samples, but the number of features becomes K, and after the dimensionality reduction, it is:

$$\tilde{X}_{n*k1} = \hat{X}_{n*m} * P_{m*k_1} \tag{6}$$

$$\tilde{Y}_{h*k_2} = \hat{Y}_t * \hat{P}_{m*k_2}. \tag{7}$$

The formula for calculating the matrix of $\check{X}, \check{Y}$ obtained by reconstructing $\hat{X}$ and $\hat{Y}_t$ is as follows:

$$\check{X} = \tilde{x}_{n*k_1} P^T_{m*k_1} = \hat{X}_{n*m} P_{m*k_1} P^T_{m*k_1} \tag{8}$$

$$\check{Y} = \tilde{Y}_{h*k_2} \hat{P}^T_{m*k_2} = \hat{Y}_t \hat{P}_{m*k_2} \hat{P}^T_{m*k_2} \tag{9}$$

where n is the number of rows in the training set, h is the number of rows in the monitoring set, m is the number of corresponding key variables, and k is the number of corresponding variables after dimension reduction.

Utilize PCA-based process monitoring methods for fault monitoring and fault trend prediction. There are two traditional statistical indices: the Hotelling-T2 index and the square prediction error (SPE) statistical index Q. Establish improved T2 statistical monitoring indices for $\check{Y}$, specifically constructing T2 statistics for $\check{Y}_{i,:}$ ($i = 1,2\ldots h$), with the calculation formula as follows:

$$T^2 = \check{Y}_{i,:} * \hat{P}_{m*k_2} * \hat{S}^{-1}_{k_2*k_2} * \hat{P}^T_{m*k_2} * \check{Y}^T_{i,:} \tag{10}$$

where $I_{mm}$ is the unit matrix and m is the dimension of the monitoring data. $\check{Y}_{i,:}$ represents the ith row vector of $\check{Y}$.

Since $\check{X}, \check{Y}$ obeys a normal distribution with a mean of zero, and the sample sampling points are independent of each other, $F_\alpha(k_1, n-k_1)$ obeys the F distribution with the first degree of freedom k1 and the second degree of freedom n-k1, and at a specific confidence degree of $1-\alpha$, usually $\alpha$ is 0.01, and the $F_\alpha(k_1, n-k_1)$ value can be obtained by looking up the table. Then the statistical control limit is determined (*Ahmad & Ahmed, 2021*), and the calculation formula is as follows:

$$T_\alpha = \frac{k_1(n^2-1)}{n(n-k_1)} F_\alpha(k_1, n-k_1) \tag{11}$$

where $1-\alpha$ is the confidence level, n is the number of samples in the training dataset, and k1 is the number of features selected after PCA.

The SPE can be used to measure the difference between the observed value and the predicted value based on the PCA model, thus identifying data points that may be anomalous. It measures the sum of squares of the differences between observed values and model predictions. Specifically, for each observation, it can be mapped to the principal component space using the PCA model, and the reconstructed predicted value can be obtained from the inverse transformation of the principal component space. Then, the difference between the observed value and the reconstructed predicted value is calculated, and the squares of the difference are added up to obtain the SPE statistic. A higher SPE value indicates a large difference between the observed value and the model prediction and may indicate an anomaly or change. The control limits for the SPE statistic were proposed by Jackson and Mud Holkar in 1979. Establish an improved SPE statistic (*Zhou, Parkj & Liu, 2016*) for monitoring $\check{Y}$, and construct the statistical index Q for $\check{Y}_{i,:}$ ($i = 1,2...h$):

$$Q = \check{Y}_{i,:} * \left( I_{m*m} - \hat{P}_{m*k_2} \hat{P}_{m*k_2}^T \right) * \check{Y}_{i,:}^T \tag{12}$$

where $\check{Y}_{i,:}$ is the row vector of row i of $\check{Y}$, $I_{m*m}$ is the identity matrix, and $\hat{P}_{m*k_2}$ is the matrix $\hat{P}_{m*k_2}$ of k eigenvectors corresponding to k eigenvalues from large to small from Eqs. (12) and (13).

Since $\check{X}, \check{Y}$ obeys a normal distribution with zero mean, and the sample sampling points are independent of each other, it has been proved in the literature (*Jacksonj & Mudholkarg, 1979*) that Eq. (18) is approximately in line with the normal distribution.

$$(Q/\theta_1)^{h_0} \sim N\left[ 1 + \frac{\theta_2 h_0(h_0-1)}{\theta_1^2}, \frac{2\theta_2 h_0^2}{\theta_1^2} \right] \tag{13}$$

$$\theta_r = \sum_{j=k1+1}^{m} \lambda_j^r, r = 1,2,3 \tag{14}$$

$$h_0 = 1 - \frac{2\theta_1\theta_3}{3\theta_2^2}. \tag{15}$$

Therefore, the approximate mean of $c$ is a normal distribution of 0, which is calculated as follows:

$$c = \frac{\theta_1[(Q/\theta_1)^{h_0} - 1 - \theta_2 h_0(h_0-1)/\theta_1^2]}{\sqrt{2\theta_2 h_0^2}}. \tag{16}$$

Then the control limit of Q is:

$$Q_\alpha = \theta_1\left[ \frac{c_\alpha h_0 \sqrt{2\theta_2}}{\theta_1} + 1 + \frac{\theta_2 h_0(h_0-1)}{\theta_1^2} \right]^{\frac{1}{h_0}}. \tag{17}$$

$c_\alpha$ is the confidence limit of the standard normal distribution, $c_\alpha$ is 0.99 confidence, $\lambda$ is the eigenvalue of the corresponding covariance matrix $R_{ta}$, and $h_0$ is the adjustment coefficient.

## Condition monitoring and fault prediction

(1) According to Eqs. (5) and (6), the $T^2$-statistic and $T^2$-statistic limit $T_\alpha$ of each sampling point were calculated.

(2) According to Eqs. (7) and (12), the SPE statistic $Q$ and the statistical limit $Q_\alpha$ of each sampling point of the monitoring data were calculated, respectively.

(3) Fault determination

If the system is operating normally, the $T^2$-value of the sample should meet $T^2 < T_\alpha$, otherwise it is considered to be faulty.

If the system is functioning normally, the $Q$-value of the sample should meet $Q < Q_\alpha$, otherwise it is considered to be faulty.

## LSTM predictive analytics

First, a training data matrix $X_t$ is constructed at each sampling point, which is $\{x(t_i)\dots x(t_{i-400})\}$ represents the first 400 steps of the current moment in the sliding window input vector, where $t$ represents the current moment.

Then, the data $X_t$ is filtered by fault-tolerant filtering (*Shaolin, Na & Wenming, 2016*) to obtain the matrix $X_t'$.

We set the telemetry fragment $\{x_{t_{k,1}}, \dots, x_{t_{k,15}}\}, (t_k = t_0 + kh, k = 1, 2, 3\dots, t_0$ is the sampling start time, h is the sampling interval) in the sliding time window $[t_k - 7, t_k + 7]$ with a window radius of 7, and sorts it from the lowest to the largest value as $\{\overline{x}_{t_{k,1}}, \dots, \overline{x}_{t_{k,15}}\}$, and constructs a quartile mean operator.

$$Q(x_{t_{k,1}}, \dots, x_{t_{k,15}}, 15) = \frac{1}{q_3 - q_1 + 1} \sum_{i=q_1}^{q_3} \overline{x}_{t_{k,i}} \tag{18}$$

where: $q_1$ and $q_3$ denote the lower and upper quartile ordinal numbers of $\{\overline{x}_{t_{k,1}}, \dots, \overline{x}_{t_{k,15}}\}$, respectively. Based on the good fault-tolerant ability of the quartile operator, a fault-tolerant smoothing filtering algorithm for sliding window quartile combinations can be constructed:

Select the appropriate window width radius H for the data slice:

$$S_k = \{x_{t_i}, t_i \in [t_k - H, t_k + H)\} \tag{19}$$

Perform quartile filtering:

$$x_{t_i}' = Q(S_k, 15). \tag{20}$$

The specific forecast calculation steps are as follows:

(1) The combination of input $X_t'$ and hidden states $h_{t-1}$ is handled by a hyperbolic tangent function tanh:

$$tanh(x) = (ex - e - x)/(ex + e - x) \tag{21}$$

$$g_t = tanh(W_g X_t' + U_g h_{t-1} + b_g). \tag{22}$$

(2) Forget the door $f_t$ will determine what information should be removed from the storage unit $C_t$ via an element-based sigmoid function

$$f_t = \sigma\left(w_f X'_t + u_f h_{t-1} + b_f\right). \tag{23}$$

(3) The input gate will determine what information will be stored in the storage cell $C_t$ via the sigmoid function:

$$i_t = \sigma\left(w_i X'_t + u_i h_{t-1} + b_i\right). \tag{24}$$

(4) Update the information in the memory unit $C_{t-1}$ by partially forgetting the information in the previous memory unit, the method is as follows:

$$C_t = f_t * C_{t-1} + i_t * g_t \tag{25}$$

where $*$ represents the element-by-element product function of the two vectors.

(5) Update the output hidden state $h_t$ based on the calculated cell state $C_t$ as follows:

$$O_t = \sigma(w_o X'_t + u_o h_{t-1} + b_o) \tag{26}$$

$$h_t = O_t * tanh(C_t). \tag{27}$$

(6) The output $y_t$ of the current moment is shown below:

$$y_t = softmax(W_y h_t + b_y). \tag{28}$$

The predicted data $y_t$ can be mathematically abstracted as $\{y(t_i), t_i = t_0 + kh_1, \text{k} = 1, 2, 3 \dots \}$, where $t_0$ and $h_1$ is the starting time of the measurement process and the sampling interval, respectively.

Through the training process, the network input weight $w_{g,i,f,o,y}$, cyclic weight $u_{g,i,f,o}$ and bias $b_{g,i,f,o,y}$ are calculated, and the trained network is used to predict the time series data. In this paper, we use the LSTM model for multi-step prediction of eigenvariables. Specifically, we used a 401-step sliding window as the input to the LSTM, and an output with a 30-step sliding window, meaning that we were predicting 30 data points in the future. To achieve this, we employ a recursive single-step prediction approach and use the predicted values to update the network state. This approach allows us to predict the next 0.5 h of data at once, thus meeting the need for real-time monitoring of future samples. In practical application, we make forecasts every 10 min to ensure the accuracy and timeliness of forecasts. This method not only improves the reliability of the monitoring judgment of each sampling point of the monitoring index, but also provides more response time for the site engineer.

## SIMULATION CALCULATION RESULT ANALYSIS

In real situations, failure data are often scarce. Safety-critical and expensive components are rarely allowed to run to failure. If the fault is not serious, the machine will usually continue to operate until repair facilities and spare parts are available. The model presented in this paper can be used to predict the impact of failures on machine operation, reduce the impact on product quality before replacement and alternatives are available, and greatly improves the problems of false detection and critical safety insecurity.

## LSTM predictive simulation analysis
### Introduction to LSTM data settings

For each fault and normal state, the compressor status is reflected in a training data set and a test data set. The LSTM contains 401 samples per training dataset and 30 samples per prediction dataset. Each sample has 12 features, the original training sample points are $401 \times 12 = 4812$, and the predicted sample points each time are $30 \times 12 = 360$. The data sampling frequency is 1 time/min. The training set is gradually updated as the monitoring time moves. A sliding window with a width of 401 before the current time is used to collect training data. The time for a prediction is 30 min, and the prediction period is once every 10 min to meet the monitoring needs. In order to ensure a good learning effect for training samples, fault-tolerant filtering and standardization are performed on each training sample obtained.

### Network structure settings

The LSTM network proposed in this paper includes an input layer, an LSTM layer, a fully connected layer and a regression layer. The shape of the input layer is a matrix with batch size 401 and features 12. The number of hidden units in the LSTM layer is 200, the initial learning rate is 0.01, and the learning rate is multiplied by the learning factor of 0.1 after 259 training sessions, which can reduce the error of the result and shorten the time. The number of training rounds is 500, and the gradient threshold is 2 to prevent gradient explosion during prediction. The output of the LSTM is then passed to the fully connected layer, which maps the patterns captured by the LSTM to the output space. Finally, through the regression layer, the network predicts the time series data and compares it with the actual data, and the accuracy of the prediction can be improved by calculating the root mean square error RMSE of the predicted and tested values.

### Parameter tuning

The choice of LSTM network in this paper is based on the structure and complexity of the data, and is continuously adjusted through experiments. For the 12-dimensional feature prediction, the input layer, LSTM layer, fully connected layer combined with regression layer are used. The role of the input layer is to provide data for subsequent layers, passing data on the time steps of the sequence data. The LSTM layer is the core of the network and is particularly suitable for processing and predicting data in time series, and the number of hidden cells set to 200 is obtained empirically, which can be optimized by setting an initial value interval and then searching according to the grid search. A setting of 200 is able to capture complex patterns while avoiding overfitting. The learning rate determines the size of each parameter update step. A larger learning rate converges quickly but may be unstable. A larger learning rate is stable but converges slowly. Therefore, this paper combines the above two advantages to achieve the rapid convergence of the model by controlling the learning rate decline period and the learning rate decline factor, and combines the Adam optimizer to realize the adjustment of the learning rate. The decline cycle and decline factor are constantly adjusted through experience and experimentation. The number of training epochs is continuously adjusted by monitoring the performance of the validation set and the validation loss curve. The threshold of gradient clipping is generally set between 0.5-5,
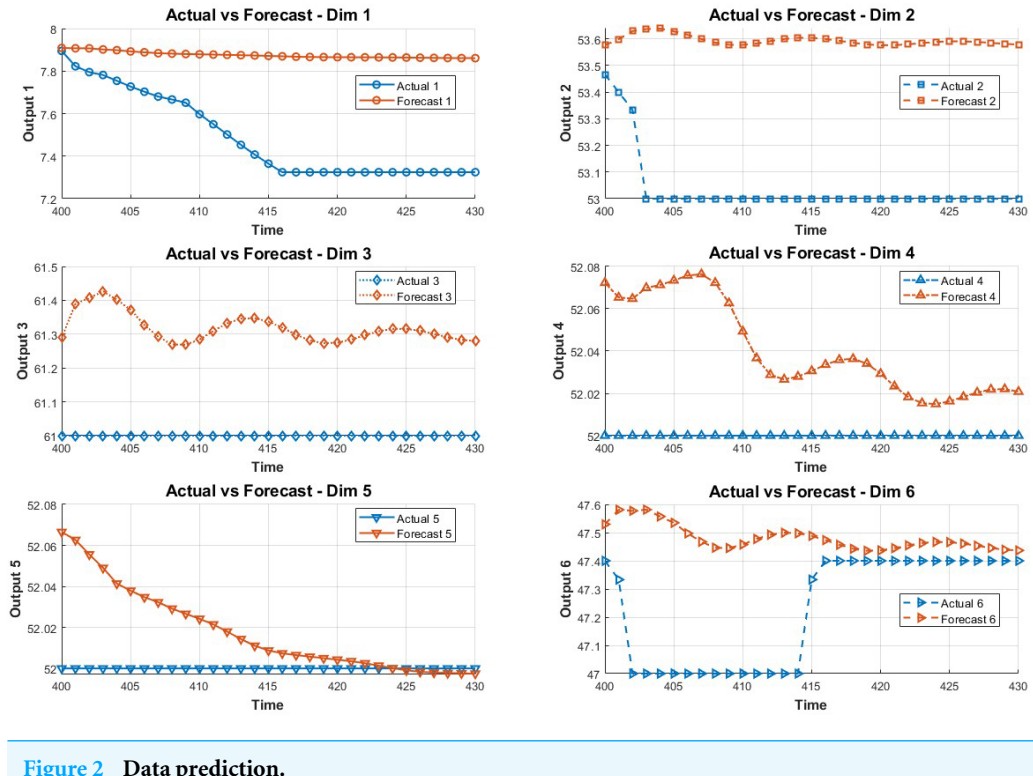

**Figure 2    Data prediction.**

which is suitable for most learning models, especially LSTM, etc., and it is common to choose 1-2, which is determined by setting the initial value and combining it with grid search. The fully connected layer maps the output of the LSTM layer to the desired output dimension. The regression layer is the output layer of the network. The function of the regression layer is to calculate the error between the predicted value and the true value to optimize the model. The reason for using a regression layer is that the task of this network is time series prediction rather than classification. By calculating the root mean square error (RMSE) between the predicted value and the test value, the accuracy of the prediction can be improved.

In order to study the influence of key variables on the system in the fault environment, the LSTM method is adopted in this paper. The method predicts the state of the next 30 data points by training the first 400 data points. The prediction results are shown in Figs. 2 and 3.

## Model evaluation

Before the fault, when the fault is detected and after the fault is detected, the predicted root mean square error, mean square error, mean absolute error, and mean absolute percentage error are as shown in Tables 1, 2, 3 and 4 below, 1–12 represents the predicted 12-dimensional variables.

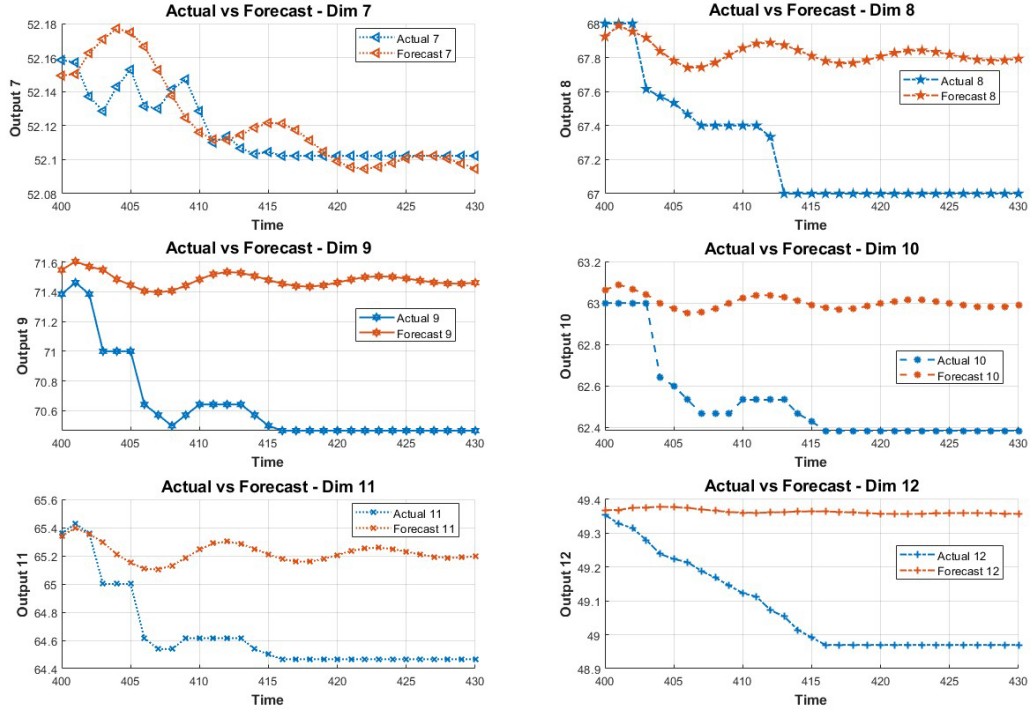

**Figure 3** The 7-12 dimensional data prediction result chart.

**Table 1 Root mean square error.**

| Variable | 1 | 2 | 3 | 4 | 5 | 6 | 7 | 8 | 9 | 10 | 11 | 12 |
|---|---|---|---|---|---|---|---|---|---|---|---|---|
| Pre-fault prediction RMSE | 0.0339 | 0.0842 | 0.1886 | 0.1069 | 0.1606 | 0.1942 | 0.2262 | 0.5568 | 0.6415 | 0.0662 | 0.6939 | 0.6148 |
| Predicting when a fault is detected RMSE | 3.9902 | 3.9562 | 3.4263 | 3.6270 | 3.1400 | 3.3887 | 1.3491 | 1.1296 | 1.3469 | 0.7478 | 2.3322 | 1.8933 |
| Predicting after detecting faults RMSE | 0.6300 | 0.7302 | 0.6324 | 0.3130 | 0.5669 | 0.6070 | 0.3756 | 0.7871 | 0.5550 | 0.2586 | 0.9502 | 0.6437 |

## Fault monitoring verification based on combining LSTM and PCA algorithm

### Introduction to fault monitoring data settings

For each failure and normal state, the compressor status is reflected in a training dataset and a test dataset. The LSTM-PCA training dataset contains 1,500 samples, and each time monitoring dataset contains the current moment, 100 past samples and 30 future samples. Each sample has 12 features, the original training sample points are $1500 \times 12 = 18000$, and the sample points monitored each time are $131 \times 12 = 1572$. The data sampling frequency is 1 time/min. The monitoring data set is gradually updated as the monitoring time moves. In order to ensure a good learning effect for the training samples, fault-tolerant filtering and standardization are performed on each training sample obtained. The constructed monitoring samples at each moment are also fault-tolerant filtered and standardized.

**Table 2  Model index evaluation results before failure.**

|       | 1      | 2      | 3      | 4      | 5      | 6      | 7      | 8      | 9      | 10     | 11     | 12     |
|-------|--------|--------|--------|--------|--------|--------|--------|--------|--------|--------|--------|--------|
| MSE   | 0.0012 | 0.0006 | 0.0254 | 0.0129 | 0.0630 | 0.0015 | 0.4621 | 0.0676 | 0.0969 | 0.0659 | 0.1661 | 0.0041 |
| MAE   | 0.0288 | 0.0200 | 0.1251 | 0.0986 | 0.1979 | 0.0315 | 0.6250 | 0.2333 | 0.2385 | 0.1997 | 0.3468 | 0.0515 |
| MAPE  | 0.2063 | 0.0308 | 0.1713 | 0.1541 | 0.3112 | 0.0543 | 0.9894 | 0.3101 | 0.3056 | 0.3008 | 0.5025 | 0.0990 |

**Table 3  Model index evaluation results when a fault is detected.**

|       | 1       | 2      | 3      | 4      | 5      | 6      | 7      | 8      | 9      | 10     | 11     | 12     |
|-------|---------|--------|--------|--------|--------|--------|--------|--------|--------|--------|--------|--------|
| MSE   | 4.9208  | 5.2977 | 5.9286 | 7.9677 | 7.3640 | 4.7036 | 5.0191 | 7.5286 | 4.8958 | 0.3368 | 1.5094 | 0.1737 |
| MAE   | 1.9185  | 1.9929 | 2.0879 | 2.4270 | 2.3355 | 1.8828 | 1.9161 | 2.4991 | 1.9485 | 0.5071 | 1.0543 | 0.3720 |
| MAPE  | 20.3747 | 3.3643 | 3.4618 | 4.5379 | 4.6401 | 3.6619 | 3.9558 | 2.4658 | 0.8114 | 1.6186 | 0.8093 | 0.7723 |

**Table 4  Model index evaluation results after a fault is detected.**

|       | 1      | 2      | 3      | 4      | 5      | 6      | 7      | 8      | 9      | 10     | 11     | 12     |
|-------|--------|--------|--------|--------|--------|--------|--------|--------|--------|--------|--------|--------|
| MSE   | 0.1971 | 0.2752 | 0.0954 | 0.0018 | 0.0138 | 0.1283 | 0.0107 | 0.5050 | 0.8914 | 0.2693 | 0.4865 | 0.0882 |
| MAE   | 0.4001 | 0.4969 | 0.7873 | 0.0316 | 0.1124 | 0.2880 | 0.0924 | 0.6247 | 0.9005 | 0.4869 | 0.6324 | 0.2697 |
| MAPE  | 5.4107 | 0.9373 | 0.4710 | 0.0704 | 0.2162 | 0.6115 | 0.1773 | 0.9311 | 1.2759 | 0.7796 | 0.9796 | 0.5502 |

### Analysis of data causes

The motor fault and its related indicators are selected for verification. After the motor fault, the faulty motor cannot maintain a relatively constant speed and power as before the fault occurred, and the motor stator temperature also gradually changes with the motor fault. They can serve as key indicators of motor failure. The motor was in a shutdown state before 10:00 on 2022/10/14, and ran normally after 10:50. An abnormality occurred around 19:30, and returned to normal around 5:30 on 10/15. The temperature of the motor stator remains unchanged when the motor is stopped, rises after the motor operates normally, drops after an abnormality occurs in the motor, and recovers after the motor recovers. The motor speed and power graphs with the motor stator temperature are shown in Figs. 4 and 5, and the motor temperature anomaly graph is shown in Fig. 6. The colors gradually transition from warm (red and yellow) to cool (blue and green).

### Analysis of the results of the calculation of monitoring indicators

This article is based on the compressor data provided by Gao Ling Station of PetroChina West-East Gas Pipeline Co., Ltd., combined with internal maintenance manuals, fault cases and the cooperation and experience of experts in related fields, to conduct monitoring and analysis of abnormal motor temperature faults. This article focuses on monitoring 12 key characteristic data such as motor power, motor stator temperature, compressor non-drive end temperature, drive end temperature, lubricating oil tank temperature, main thrust bearing temperature and auxiliary thrust bearing temperature. The causes of the failure and their solutions are summarized as shown in Table 5.

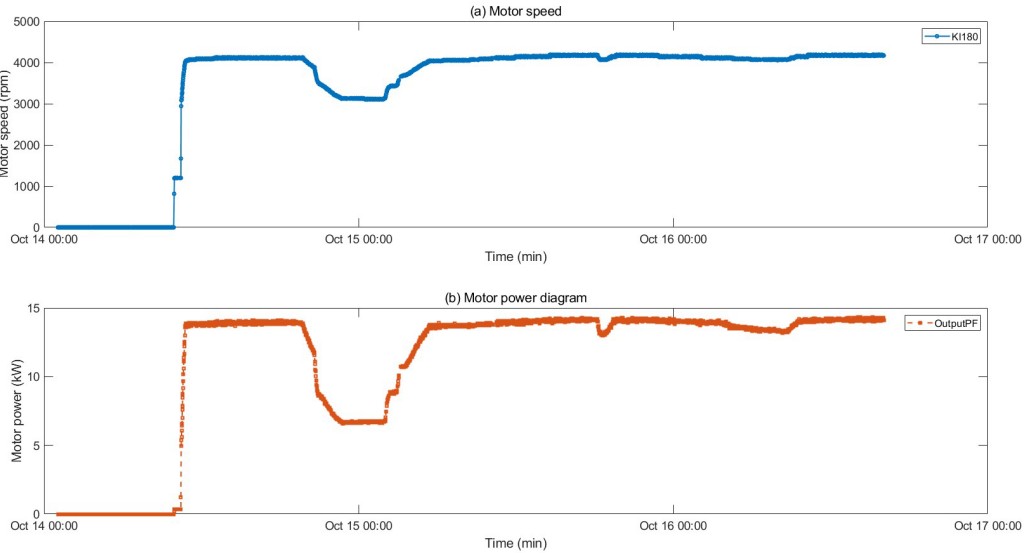

**Figure 4** **Trend of speed and power changes before and after motor failure.**

The monitoring method proposed in this article not only focuses on the impact of a single feature on faults, but also further improves the accuracy of fault detection through comprehensive analysis of multi-dimensional features. Each sampling point contains a combination of the first 100 historical data and the last 30 future data, which greatly improves the accuracy and reliability of monitoring.

The fault is detected at 30 points of fault degradation. At the monitoring moment If the first 10 data are all abnormal, it is considered that an abnormality has occurred. Therefore, when a fault is detected, this paper considers that a fault has occurred. The fault monitoring results calculated from Eqs. (5), (6), (7) and (12) are shown in Fig. 7.

### Comparison of improved methods

The monitoring effect before the algorithm improvement is shown in Fig. 8. Prior to the improvement, each point only contained the monitoring effect of the current point, leading to false detections, false alarms, and insufficient information content.

After the improvement, each sampling point now includes 100 historical data points prior to the sample and 30 future data points following it, greatly enhancing the reliability and accuracy of the monitoring alarms. If the 10 data points before the monitoring moment are all abnormal, an anomaly is considered to have occurred, and the forecast results for the time after the current moment are provided. The normal monitoring results for each sampling point are shown in Fig. 9, thereby allowing the previously mentioned false detections to be monitored correctly.

## CONCLUSIONS

Based on the condition monitoring data of industrial centrifugal compressors, this paper tests the fault detection and identification ability of PCA. At the same time, for the first

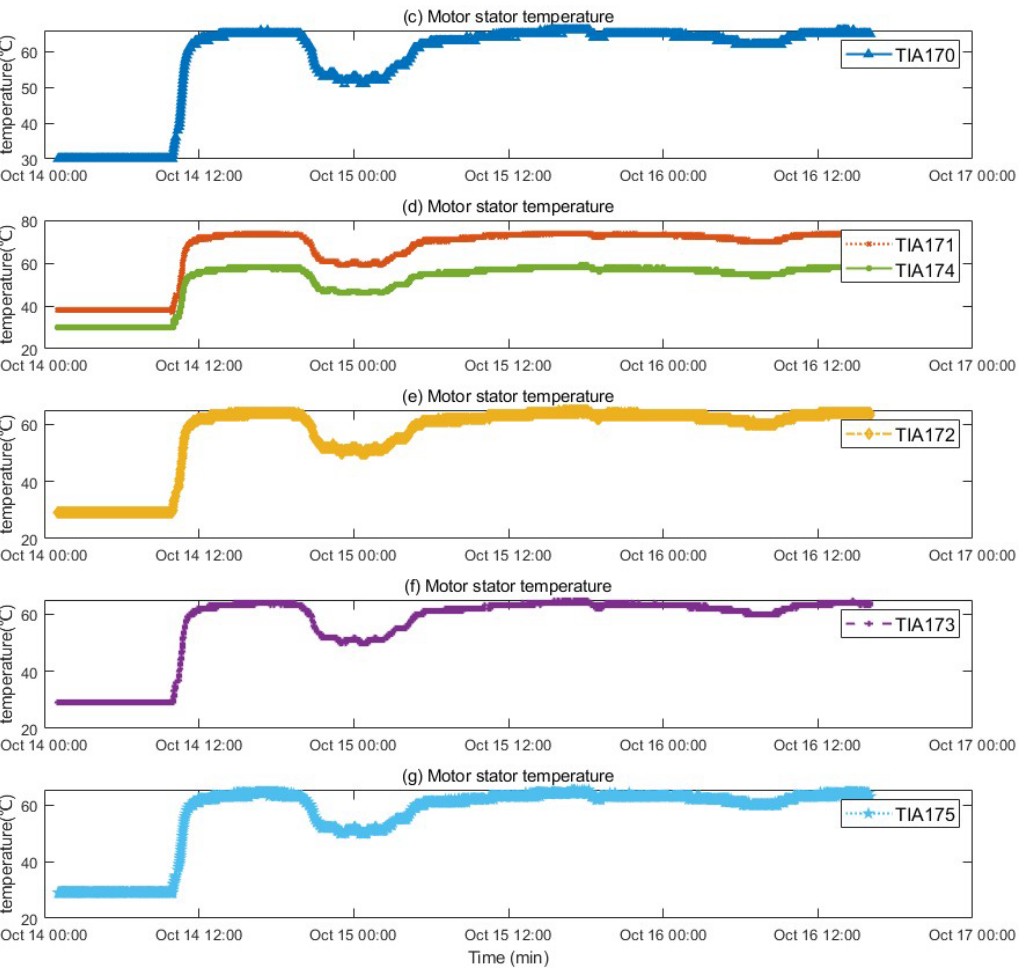

**Figure 5** Trend of stator temperature change before and after motor failure.

time, PCA and LSTM are combined to predict the behavior of the system under fault conditions, and combined with the statistics and SPE statistics health indicators, the fault is successfully detected in a short detection time and the reaction time is given.

This combined method can greatly improve the following problems: Firstly, critical safety is not necessarily safe. By means of future data prediction combined with the mentioned monitoring algorithm, this article achieves fault prediction and greatly improves the situation where the fault trend develops rapidly and it is too late to repair the system. Secondly, the existence of outliers in the monitoring data causes frequent false alarms in the system. Due to the harsh working environment of the compressor and a large amount of disturbance, the monitoring sensors are prone to outliers, thus reaching the system's threshold value. This paper greatly reduces the impact of environmental interference by comprehensively considering the impact of multi-dimensional features on a single fault, and greatly improves the performance by constructing at each monitoring point the 100 historical data before the current sampling point and the 30 future data after the sampling point. It eliminates the influence of outliers and improves the accuracy and reliability of monitoring.

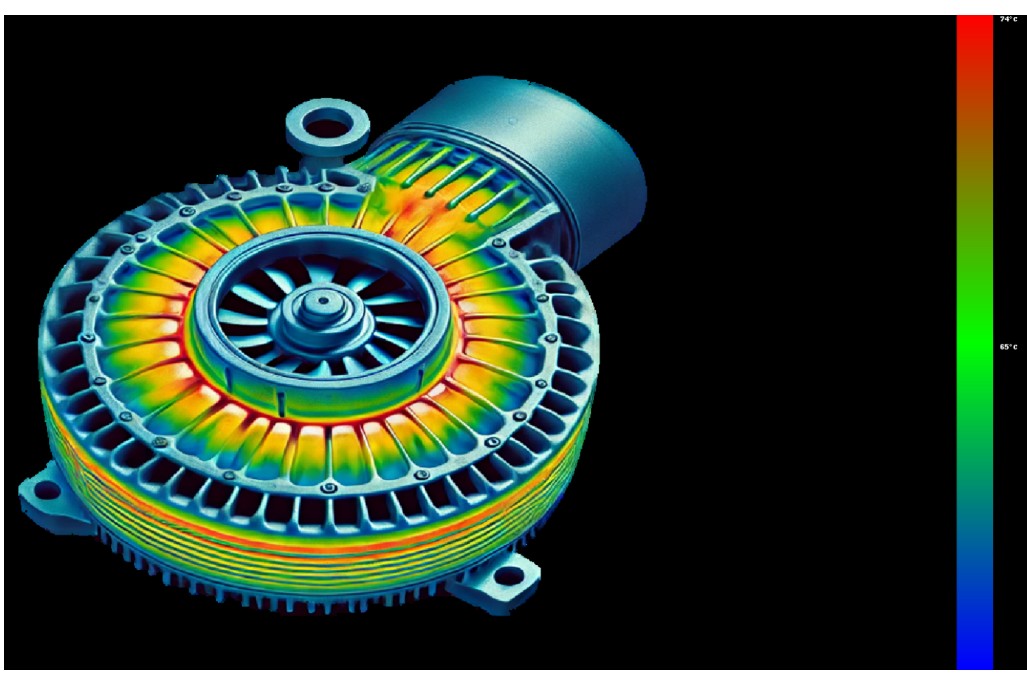

**Figure 6  Motor temperature abnormality diagram.**

**Table 5  Motor faults and solutions.**

| Cause of failure | Solutions |
| --- | --- |
| There is something wrong with the pump | The main oil pump does not start: there is no working medium, notify the responsible department. The auxiliary oil pump does not start when the oil pressure drops: electrical failure, electrical failure of the pump automatic equipment (notify the responsible department) |
| Oil pipeline leak | Leakage at flange connection: seal must be replaced. Oil line rupture hazard: Fire hazard due to contact with hot parts |
| Cooler, filter or strainer is dirty | Conversion cooler, filter cleaning coarse filter |
| Defective oil pressure balancing valve or pressure reducing valve | Check valve and replace if necessary |
| Oil pressure too low | The pump is defective, the oil pipeline is leaking, the cooler, filter or strainer is dirty, the oil pressure balance valve or pressure reducing valve is defective |
| The oil level in the high tank is too low | Add enough oil |
| Seal oil pressure difference is too low | Adjust control valve |
| Oil temperature is too low | turn on heater |
| Fault | Motor failure |

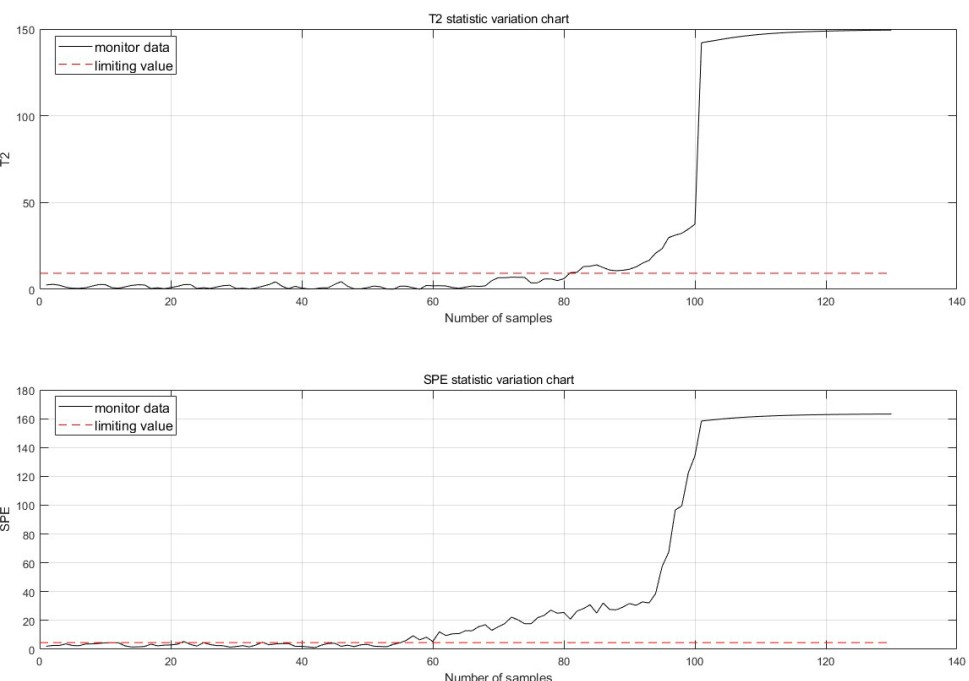

**Figure 7  Fault monitoring.**

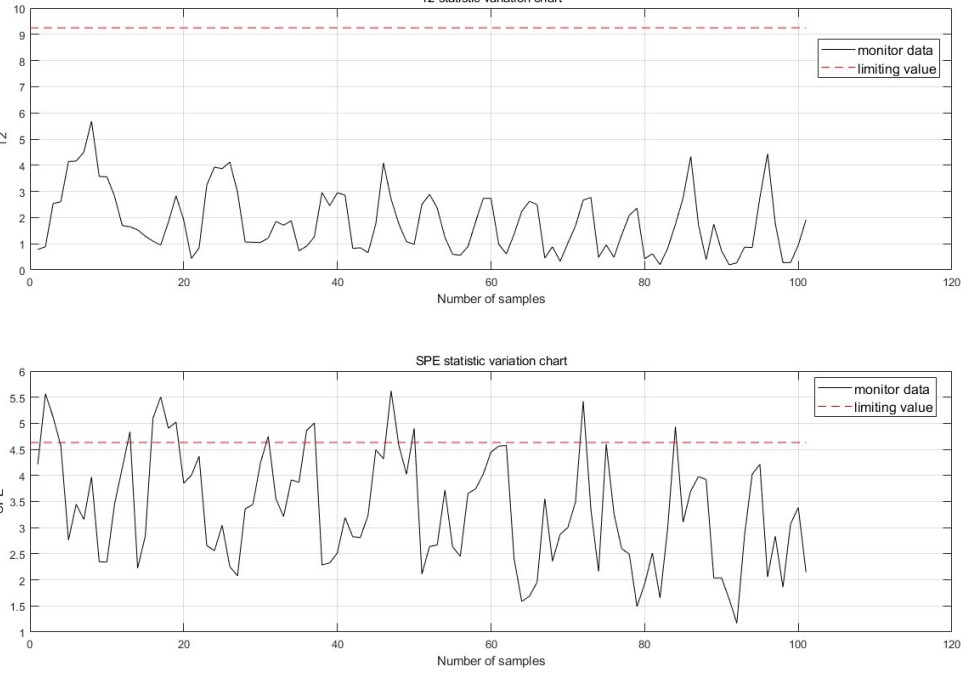

**Figure 8  Monitoring chart before improvement.**

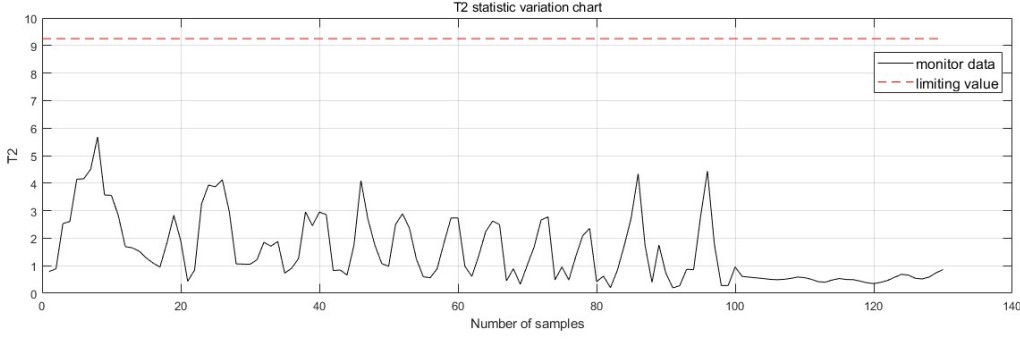

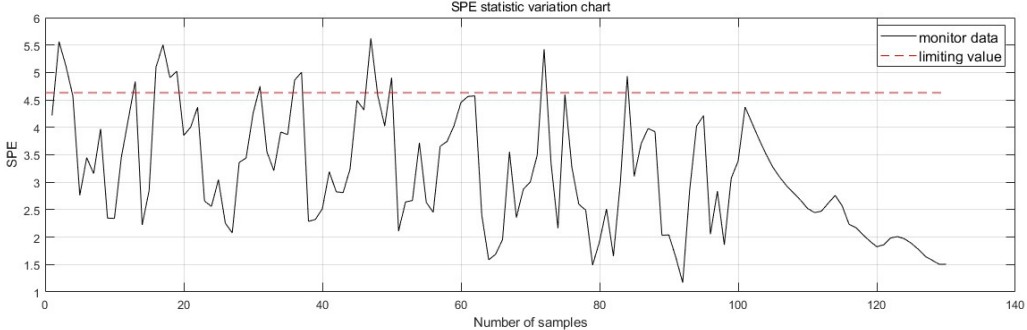

**Figure 9** **Single point monitoring map.**

## Funding
This work was supported by the Natural Science Foundation of China (62373115); Nature and Science Foundation of Guangdong Province (2023A1515012341,2024A1515010870). The funders had no role in study design, data collection and analysis, decision to publish, or preparation of the manuscript.

## Grant Disclosures
The following grant information was disclosed by the authors:
The Natural Science Foundation of China: 62373115.
Nature and Science Foundation of Guangdong Province: 2023A1515012341, 2024A1515010870.

## Competing Interests
The authors declare there are no competing interests.

## Author Contributions
- Yuan Wang conceived and designed the experiments, performed the experiments, analyzed the data, performed the computation work, prepared figures and/or tables, and approved the final draft.

- Shaolin Hu conceived and designed the experiments, analyzed the data, authored or reviewed drafts of the article, and approved the final draft.

## Data Availability

The raw measurements are available in the Supplementary Files 1.

## Supplemental Information

Supplemental information for this article can be found online at http://dx.doi.org/10.7717/peerj-cs.2433#supplemental-information.

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
