# Peer review of "State monitoring and fault prediction of centrifugal compressors based on long short–term memory and principal component analysis (LSTM-PCA)"

_PeerJ Computer Science, doi:10.7717/peerj-cs.2433_

## Round 0.1 · original submission · Major Revisions

The reviewers highlighted several critical issues in the manuscript, including insufficient detail in the experimental method, a lack of in-depth data analysis, and an inadequate introduction to the research background, which fails to fully explain the innovation and importance of this study as well as overlooking recent research in centrifugal compressor fault diagnosis. It is also essential to enhance methodological detail and perform a more comprehensive data analysis.

Reviewer 1 ·

Basic reporting

1. The description of the experimental method is not detailed enough and lacks a detailed description of the key steps and parameters.
2. Lack of in-depth analysis of the data, failing to provide sufficient evidence to support the research conclusions.
3、Insufficient introduction of the research background, failing to fully explain the innovation and importance of the study.
4、The writing quality of the manuscript is poor, with language errors and unclear expressions that affect reading comprehension.
5, The manuscript has limited practical application value and fails to provide a substantial contribution to the development of the field.
6, The results section is repetitive and redundant, and lacks in-depth discussion of the main findings.

Experimental design

no comment

Validity of the findings

no comment

Additional comments

no comment

·

Basic reporting

The writing is strong overall. However, the introduction and background sections could be enhanced. The proposed concept is promising but requires further development to be truly compelling. Employing deep learning with composite concepts, such as LSTM and PCA, for anomaly detection is not groundbreaking. The figures need improvement. For instance, Figure 4's title should accurately reflect its content rather than simply stating the statistical method used. Figures 5, 6, and 7 are highly correlated and should be combined into a single figure with subfigures labeled (a), (b), and (c) for better visualization.

Experimental design

The research question to use AI to monitor centrifugal compressors to improve, ensure safe and efficient operations is well defined and conducted. Data and statistical technique are suficiently provided and described. This work is good.

Validity of the findings

The data have been provided. They are robust and sound. Conclusion is precise and Ok.

·

Basic reporting

This paper presents a dynamic process monitoring method for centrifugal compressors based on long short-term memory and principal component analysis. This method constructs a sliding window for monitoring at each sampling point and uses LSTM to predict 30 future data points. While this paper presents certain results, this paper is required Major Revisions to be a good journal paper. Some comments and suggestions are given below:
1) Literature review is poorly conducted. The authors have overlooked the research progress in fault diagnosis of centrifugal compressors, while most of the references cited in the paper are either machine learning methods or PCA method.
2) The authors are suggested to read some recent papers in fault diagnosis of centrifugal compressors and understand the current status of the research. If the references in centrifugal compressors could not be found, then authors could search for Intelligent fault diagnosis of a reciprocating compressor. These two types of compressors have been widely used in petroleum industry.
3) A number of figures/photos should be provided to clearly show the fault types.
4) The quality of the figures should be improved, to make them look nicely.

Experimental design

see above

Validity of the findings

see above

Additional comments

see above

---

## Round 0.2 · Minor Revisions

The revisions made by the authors are appreciated. However, the reviewer would still like the paper to be improved further before it is fit for publication.

Reviewer 1 ·

Basic reporting

After revisions, the overall quality of the manuscript has significantly improved. However, several issues still need further attention:
1. There are formatting errors in the equations and variables throughout the manuscript. Please review and correct them thoroughly.
2. Figure 1 should be redrawn to further enhance its quality and aesthetics.
3. The vertical axis scale in Figure 2 is too large, making it difficult to clearly observe the information presented.
4. Figures 3-6 and Figure 10 should be converted into tables.
5. The quality of Figures 7-8 and Figures 11-13 needs to be improved, and different curves should be clearly distinguished.
6. Important parameters should be added to Figure 9.
7. The number of references should be increased to provide a more comprehensive review of the current research landscape

Experimental design

No comment.

Validity of the findings

No comment.

Additional comments

No comment.

---

## Round 0.3 · accepted · Accept

The authors have addressed the comments made by the reviewer and the paper is now fit for publication.